# ATTEND TO CONTEXT FOR REFINING EMBEDDINGS IN DEEP METRIC LEARNING

## ABSTRACT

The primary objective of deep metric learning (DML) is to find an effective embedding function that can map an image to a vector in the latent space. The quality of this embedding is typically evaluated by ensuring that similar images are placed close to each other. However, the evaluation step, which involves finding the sample and its neighbors and determining which neighbors share the same label, is often overlooked in the current literature on DML, where most of the focus is placed on training the embedding function. To address this issue, we propose a mechanism that leverages the statistics of the nearest neighbors of a sample. Our approach utilizes cross-attention to learn meaningful information from other samples, thereby improving the local embedding of the image. This method can be easily incorporated into DML approaches at a negligible additional cost during inference. Experimental results on various standard DML benchmark datasets demonstrate that our approach outperforms the state of the art.

## 1 INTRODUCTION

Deep metric learning (DML) is a powerful technique to learn compact image representations that can generalize well so that transfer to previously unseen data distributions. The goal of DML is to not only express semantic similarities between training samples but also to transfer them to unseen classes. Thus, the primary objective is to find an embedding function that can map images to their corresponding locations in an embedding space where the semantic similarity is implicitly captured by the distance between samples. By doing so, we can ensure that positive images with the same label are located close to each other while images with different labels are located far apart. This is a crucial step in improving the performance of various visual perception tasks, such as image retrieval Sohn (2016); Wu et al. (2017); Roth et al. (2019); Jacob et al. (2019), clustering Hershey et al. (2016); Schroff et al. (2015a), and face/person identification Schroff et al. (2015b); Hu et al. (2014); Liu et al. (2017); Deng et al. (2019).

Conventional deep metric learning approaches typically process each image independently of others. This means that each image is fed into the neural network individually, and the model generates an embedding in a latent space. However, during the evaluation stage, we retrieve the nearest neighbors for each sample and check if they share the same label as the query image. Thus, we judge the model on its ability to properly arrange points in the neighborhood.

Additionally, large discrepancy between the train and test sets often characterizes benchmark datasets for transfer learning common for DML. In this challenging non-i.i.d. setting, computing an embedding only for an individual query sample makes it difficult to adjust to the domain shift. While there are approaches that can train in unsupervised way on the test set Sun et al. (2020) this is not always possible. However, considering related samples when embedding a query point, we could better discover and compensate for global domain shifts and focus on relations between samples in the neighborhood. But what information from which other sample is meaningful for improving the embedding of a query point? Our proposal is to learn how to extract informative characteristics from the other samples in the neighborhood and combine this information with the query sample. Similar to the approach described in the perceiver paper Jaegle et al. (2021), our method utilizes cross-attention blocks to analyze the relationships between the sample and its neighbors.

We observe that sampling from the neighborhood of the query prioritizes meaningful samples. Here the neighborhood is defined by whatever initial DML feature extractor is used prior to the embedding

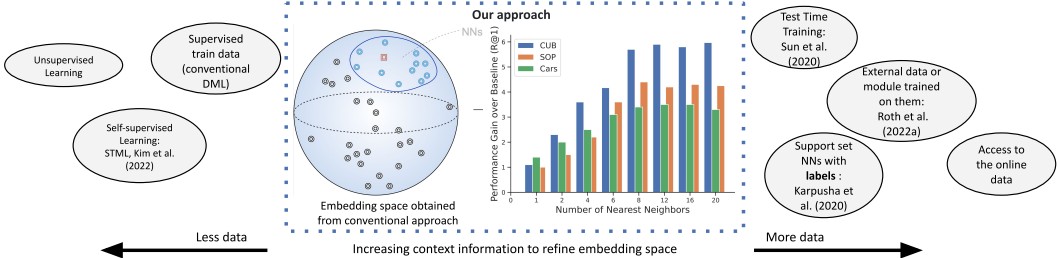

Figure 1: Our approach falls between the supervised and unsupervised scenarios. While there are numerous works on conventional DML, which involve labeled training datasets and evaluation on separate test datasets, there are also approaches on the right side of our approach that have access to the test set in either a supervised or unsupervised form. These approaches can optimize their models based on this test set in various ways. We propose fixing the evaluation function in a way that allows our model to have access to the neighborhood of points during the evaluation stage. This will enable our model to readjust the embedding of the query sample accordingly. Notably, our model consistently outperforms the baselines when increasing the number of neighbors each sample can contextualize.

that we want to learn. By focusing on relationships within the data, our approach is able to extract more information from the available data. This ultimately results in enhanced embeddings and, as a result, improved retrieval of nearest neighbors with the correct label.

Furthermore, our approach can be applied on top of the initial features or even the final embeddings computed with any existing method, regardless of its complexity or the nature of the data being analyzed. Our approach can better reflect the underlying structure of the data in the neighborhood of a sample and henceforth improve the retrieval performance by a significant margin. To put it simple, our approach can be seen as a mechanism that allows different samples to communicate with each other and improve their embeddings based on the relationships that exist between them. This is particularly useful when dealing with complex datasets, where the initial embeddings may not fully capture the nuances and intricacies of the underlying structure.

In summary, we have designed an easy-to-apply approach that can refine precomputed embeddings obtained with any vanilla DML approach. Our approach has the following characteristics:

- easy to incorporate into existing DML approaches
- adds negligible computation overhead at inference time
- targets the problem of a distribution shift in DML
- significantly outperforms state-of-the-art approaches on all main benchmark datasets used in DML
- breaks the assumption of conventional approaches that images exist independently from each other in the embedding space

## 2 RELATED WORK

**Deep Metric Learning:** DML is a prominent research field focusing on similarity learning and its various applications such as image retrieval and search Sohn (2016); Wu et al. (2017); Roth et al. (2019); Jacob et al. (2019), face recognition Schroff et al. (2015b); Hu et al. (2014); Liu et al. (2017); Deng et al. (2019), zero-shot learning Bautista et al. (2016); Sanakoyeu et al. (2018); Büchler et al. (2018), and clustering Hershey et al. (2016); Schroff et al. (2015a). The primary objective of DML is to optimize the projections of individual images into an expressive embedding space so that similarity relations between the images are captured by a given distance function. To achieve this goal, numerous approaches for learning have been proposed, including surrogate ranking tasks over tuples of images, ranging from simple pairs Hadsell et al. (2006) and triplets Wu et al. (2017); Schroff et al. (2015b); Wang et al. (2017); Deng et al. (2019) to higher-order quadruplets Chen et al. (2017) and more generic n-tuples Sohn (2016); Oh Song et al. (2016); Hermans et al. (2017); Wang et al. (2019). The number of different combinations of tuples usually grows exponentially, but most of them are uninformative. To tackle this issue, another stream of works in DML is focusing on

various sampling strategies for mining informative tuples, including Wu et al. (2017); Schroff et al. (2015b); Xuan et al. (2020); Ge (2018); Harwood et al. (2017); Roth et al. (2020). To circumvent the sampling issue, some proxy-based Goldberger et al. (2005); Movshovitz-Attias et al. (2017); Kim et al. (2020); Teh et al. (2020); Qian et al. (2019) or classification-based Deng et al. (2019); Zhai & Wu (2018) methods are also proposed. Apart from these basic formulations, diverse extensions, for instance, generating synthetic training samples Duan et al. (2018); Lin et al. (2018); Zheng et al. (2019); Gu et al. (2021); Ko & Gu (2020), teacher-student approach Roth et al. (2021), leveraging additional tasks and ensemble learning Opitz et al. (2017; 2018); Sanakoyeu et al. (2021); Roth et al. (2019); Milbich et al. (2020); Kim et al. (2018), are proven to be effectively enhancing different capabilities of DML models.

We argue that computing an embedding only locally for an individual query sample makes it difficult to adjust the domain shift problem existing in many DML benchmark datasets and real-world scenarios Milbich et al. (2021). In our work, we break with this paradigm by improving the embeddings based on their surrounding context.

**Utilizing contextual information:** Intra-Batch Seidenschwarz et al. (2021) proposed to utilize the contextual information contained in a training mini-batch by means of message passing networks. However, this cannot be easily applied to test time, as the method is highly sensitive to the construction of the mini-batch (see discussion in their supplementary). STML Kim et al. (2022) proposed to use contextualized semantic similarity by considering the overlap of the k-reciprocal nearest neighbors of data in the embedding space for self-supervised metric learning, where the author perform nearest neighbors searching to construct the training batch. To further exploit available datasets, Frosst et al. (2019) introduces a trust score that measures the conformance between the classier and k-nearest neighbors on a set of examples with known labels. Similarly, Karpusha et al. (2020) calibrates their prediction based on nearest neighbors with labels in the held-out validation dataset to improve generalization and robustness of deep metric learning models. Meanwhile, Roth et al. (2022a) explores the contextual information (represented by top-$k$ ImageNet pseudo-labels) in the language domain by guiding the training process through KL-divergence between image and text similarities defined by CLIP Radford et al. (2021), which is trained on 400 million image-text pairs.

On the other hand, our approach does not rely on test-batch construction, usage of data labels, or module potentially posing a data leakage problem, but aims at leveraging the context information contained in the very test set neighborhood and improving the embedding based on it. Additionally, this also offers us a tool to diagnose how the image representation changes with its neighbors served as the context. We showed related experiments in sec. 4.

**Attention mechanisms:** transformer architecture has revolutionized the field of natural language processing Vaswani et al. (2017) and been gaining more and more focus in the vision domain as well Dosovitskiy et al. (2021). It enables the model to attend to specific parts of their input Jaderberg et al. (2015), feature representations Vaswani et al. (2017) or even output Jaegle et al. (2022). Of particular relevance to DML, the model proposed in El-Nouby et al. (2021) simply replaced the feature extractor with Vision Transfomer (ViT) and was trained with DML objectives, which led to significant improvement over conventional backbones. In Seidenschwarz et al. (2021) a message-passing network based on attentional principles was used to incorporate global structure of embeddings within a mini-batch during training. However the useful information is highly constrained by the randomly sampled mini-batch.

Another attention mechanism that has proven to be a flexible in relating two arbitrary data representations is cross attention Jaegle et al. (2021; 2022). The model is capable of scaling to hundreds of thousands of inputs and leverages an asymmetric attention mechanism to distill inputs into a tight bottleneck. In our work, we propose to utilize this flexibility to exchange information among neighboring learned embeddings, thus refining the data representations.

## 3 APPROACH

The main goal of DML is to learn a similarity measure between any arbitrary pair of samples. The measurement is defined as a similarity function $s(I_i, I_j)$ over images $I_i, I_j \in \mathcal{I}$ parameterized by a backbone network $E$ extracting features $E(I_i)$ and a function $\phi$ projecting data into the final

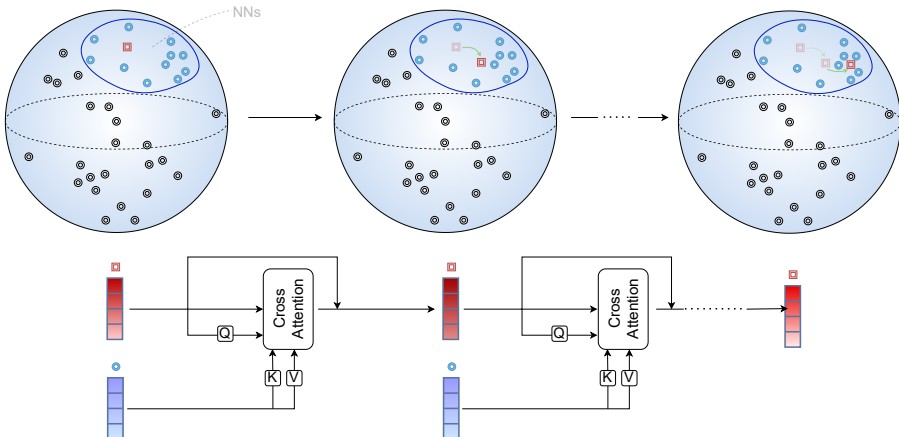

Figure 2: *Approach overview.* Our approach works by iteratively improving an initial embedding $e_q^0$ to its final embedding $e_q^T$. For this, an initial neighborhood of close embeddings is gathered. The neighbors are then used to update the current embedding $e_q^t$ by predicting missing, residual information with cross-attention.

embedding $e_i = \phi(E(I_i))$. The embedding $e_i$ is usually normalized. In this learned metric space the semantic similarity is usually represented by cosine similarity between samples.

With the similarity defined, the training loss $\mathcal{L}_{dml}$ for updating $E$ and $\phi$ usually involves solving a ranking problem between samples Wang et al. (2019); Wu et al. (2017); Qian et al. (2019) or between samples and proxies Gu et al. (2021); Kim et al. (2020); Roth et al. (2022b). For instance, in the simplest case, we have a triplet of samples, i.e. anchor $x_a$, positive $x_p$ and negative $x_n$ with their ground truth label $y_a = y_p \neq y_n$, and the model learns to ensure the $d_\phi(x_a, x_n)$ is larger than $d_\phi(x_a, x_p)$ by a certain margin.

As the reader can notice, the embeddings $e_i$ are computed independently from each other. This results in "approximately" good embeddings. That means that for any query embedding $e_q$ we can compute the set of nearest neighbors from the set of all other embeddings $e_i$ with the similarity measure $s(\cdot, \cdot)$. And this neighborhood always has a correct retrieval sample which may not be the nearest neighbor to our query sample

Henceforth, our primary goal is to develop a model that can take a query embedding $e_q$ and its neighborhood $NN(e_q)$ of independently embedded images and establish connection between them. Now we want to refine positioning of those points relative to one another. Since conventional approaches often achieve a good initial embedding, it makes sense to focus only on a small set of neighboring images – thus embedding size can have fairly small size.

With this example in mind we aim to design a model that can take the embedding $e_q$ of a query image $I_q$, along with contextual information from its nearest neighbors, and aggregate and process it to refine the initial embedding $e_q$.

Attention is a suitable mechanism for establishing correspondence between different objects. Specifically, we want to learn an effective way to exchange information between the embedding $e_q \in \mathbb{R}^d$ of a query image $I_q$ and the context set of neighboring embeddings $C_q := NN(e_q) \in \mathbb{R}^{k \times d}$ of $k$ images. The initial formulation suggested by Vaswani et al. (2017) can be applied to our data and can be formulated as follows:

$$\text{Attn}(e_q, C_q) := \text{softmax}\left(\frac{e_q C_q^\top}{\sqrt{d}}\right) \in \mathbb{R}^{1 \times k}. \tag{1}$$

This expression represents similarity between the query image and each of its $k$ neighbors. If we use it directly to aggregate information, we obtain a weighted sum of the embeddings of the nearest neighbors. But, we want to extract diverse information from the nearest neighbors $C_q = NN(e_q)$ to improve the embedding of the query image, instead of averaging the embeddings. Another limiting

factor is that using Eq. 1, similarities stay the same as they were in the original embedding space. However, we may want to focus on certain aspects of the embeddings more than others. This can only be achieved by projecting the original space into a different subspace using, for example, linear projection layers. This mechanism is implemented in the cross-attention layer, which takes three inputs: queries $q$, keys $k$, and values $v$. The layer then projects them onto a new space with projection layers $Q$, $K$, and $V$, and then assembles them back together:

$$\mathrm{CA}(q, k, v) := \mathrm{softmax}\left(\frac{Q(q)\,K(k)^{\top}}{\sqrt{d}}\right) V(v). \tag{2}$$

Using the cross-attention mechanism, we can assign higher weights to meaningful elements than to others. Given these attentions, we can merge the neighbor information with the query image embedding $e_q$. We treat this new information (output of the cross-attention block) as the residual to the initial information contained in the query image embedding $e_q$. Making the output residual to the original input stabilizes the training. This process can be repeated multiple times to iteratively improve the embedding of a query image using information about its nearest neighbors

$$e_q^t := \mathrm{CA}^t(e_q^{t-1}, C_q, C_q) + e_q^{t-1}, \ t \in \{1, .., T\}. \tag{3}$$

This design is needed, such that different cross-attention blocks will focus on different details of the neighbors $C_q$. The iterative updating process is depicted in Fig. 2.

To train the weights of all our cross-attention blocks between neighbor information and the iteratively refined embedding $e_q$, we need a loss function. We opt for a simple multi-similarity loss, which uses ground truth labels provided by the dataset. Given ground-truth labels for each sample $e_i^T$ we can establish a subset of positives $\mathcal{P}_i$ and its negatives $\mathcal{N}_i$ in a minibatch of $b$ samples. Now we can compute similarities between $e_i^T$ and its positives and $e_i^T$ and negatives. Those are being fed to the logexp function and summed together using hyperparameter $\alpha$ and $\beta$ for balancing fraction of positives and negatives in a batch:

$$\mathcal{L} := \frac{1}{b}\sum_{q=1}^{b}\left(\frac{1}{\alpha}\log\left[\sum_{k\in\mathcal{P}_q}\exp^{-\alpha((e_q^T)^{\top}e_k^T - \lambda)}\right] + \frac{1}{\beta}\log\left[\sum_{k\in\mathcal{N}_q}\exp^{\beta((e_q^T)^{\top}e_k^T - \lambda)}\right]\right). \tag{4}$$

To facilitate effective training we need to compute once all the initial embeddings for the whole dataset using pretrained networks from conventional approaches. Now, since our method operates on the neighborhoods of points it make sense to compute and store them at the very beginning of training. In short form training and inference is formulated in Alg.1 and Alg.2.

## 4 Experiments

### 4.1 Datasets Description

There are three main datasets that are used to benchmark performance of the DML methods. Following conventional approach we train our models on the train split and evaluate on the test split: *CUB200-2011* Wah et al. (2011) contain 200 classes containing in total 11, 788 images of birds. Training set contains first 100 classes with 5, 864 images and test split contains next 100 classes with 5, 924 images. *CARS196* Krause et al. (2013) has 16, 185 images across 196 different car brands. Train split contains first 98 classes totalling 8, 054 images. The remaining 98 classes with 8, 131 images

Table 1: **Additional metrics** suggested in Musgrave et al. (2020) were computed, R-Precision(RP), Mean Average Precision at R (MAP@R) and Mean Average Precision at 1000, to provide extra insights on the performance of our method.

| Datasets | RP | mAP@R | mAP @1000 |
|----------|------|-------|-----------|
| CUB | 43.2 | 33.9 | 42.8 |
| CARS | 42.7 | 34.1 | 40.4 |
| SOP | 55.7 | 55.4 | 48.1 |

are used for testing. *Stanford Online Products (SOP)* Oh Song et al. (2016) consists of images of eBay items for sale uploaded by both customers and stores. This dataset contains much bigger

Table 2: **Comparison to the state-of-the-art methods** on *CUB200-2011*, *CARS196*, and *and SOP*. 'BB' denote the backbone architecture being used ('R50'=ResNet50 He et al. (2016), 'BNI'=BN-InceptionNet Szegedy et al. (2015)). We report our results using both 512-dimensional and 2048-dimensional space.

| Method | BB | CUB200-2011 | | | CARS196 | | | SOP | | |
|---|---|---|---|---|---|---|---|---|---|---|
| | | R@1 | R@2 | NMI | R@1 | R@2 | NMI | R@1 | R@10 | NMI |
| Margin[128]Wu et al. (2017) | R50 | 63.6 | 74.4 | 69.0 | 79.6 | 86.5 | 69.1 | 72.7 | 86.2 | 90.7 |
| Multi-Sim[512]Wang et al. (2019) | BNI | 65.7 | 77.0 | 68.8 | 84.1 | 90.4 | 70.6 | 78.2 | 90.5 | 89.8 |
| MIC[128]Roth et al. (2019) | R50 | 66.1 | 76.8 | 69.7 | 82.6 | 89.1 | 68.4 | 77.2 | 89.4 | 90.0 |
| HORDE[512]Jacob et al. (2019) | BNI | 66.3 | 76.7 | - | 83.9 | 90.3 | - | 80.1 | 91.3 | - |
| Softtriple[512]Qian et al. (2019) | BNI | 65.4 | 76.4 | 69.3 | 84.5 | 90.7 | 70.1 | 78.3 | 90.3 | 92.0 |
| XBM[128] Wang et al. (2020) | BNI | 65.8 | 75.9 | - | 82.0 | 88.7 | - | 80.6 | 91.6 | - |
| PADS[128] Roth et al. (2020) | R50 | 67.3 | 78.0 | 69.9 | 83.5 | 89.7 | 68.8 | 76.5 | 89.0 | 89.9 |
| GroupLoss[1024] Elezi et al. (2020) | BNI | 65.5 | 77.0 | 69.0 | 85.6 | 91.2 | 72.7 | 75.1 | 87.5 | 90.8 |
| DIML[512]Zhao et al. (2021) | R50 | 67.9 | - | - | 87.0 | - | - | 78.5 | - | - |
| ProxyAnchor[512] Kim et al. (2020) | BNI | 68.4 | 79.2 | - | 86.1 | 91.7 | - | 79.1 | 90.8 | - |
| D&C[512]Sanakoyeu et al. (2021) | R50 | 68.2 | - | 69.5 | 87.8 | - | 70.7 | 79.8 | - | 89.7 |
| SynProxy[512] Gu et al. (2021) | R50 | 69.2 | 79.5 | - | 86.9 | 92.4 | - | 79.8 | 90.9 | - |
| DiVA[512] Milbich et al. (2020) | R50 | 69.2 | 79.3 | 71.4 | 87.6 | 92.9 | 72.2 | 79.6 | 91.2 | 90.6 |
| Intra-Batch[512] Seidenschwarz et al. (2021) | R50 | 70.3 | 80.3 | 74.0 | 88.1 | 93.3 | 74.8 | 81.4 | 91.3 | 92.6 |
| S2D2[512] Roth et al. (2021) | R50 | 70.1 | 79.7 | 71.6 | 89.5 | 93.9 | 72.9 | 80.0 | 91.4 | 90.8 |
| Multi-Sim+PLG[512] Roth et al. (2022a) | R50 | 70.0 | 79.5 | 70.8 | 87.3 | 92.5 | 73.2 | 79.1 | 91.1 | 90.1 |
| **Ours**[512] | R50 | **73.2** | **82.8** | **75.8** | **90.9** | **94.3** | **76.0** | **81.8** | **92.3** | **93.0** |
| **Ours**[2048] | R50 | **74.8** | **83.8** | **76.9** | **91.4** | **94.5** | **77.6** | **82.3** | **92.7** | **93.4** |

number of images $120,053$ spread across $22,634$ product classes. Thus, this dataset is not only bigger compared to the previous two, but also has on average fewer images per class. Training split contains $59,951$ images of $11,318$ different products. Test split contains remaining $11,316$ products of $60,502$ images in total.

## 4.2 COMPARISON TO STATE-OF-THE-ART

We evaluate our approach on aforementioned three standard benchmark datasets and compare it to the other state-of-the-art models utilizing conventional ResNet-50 He et al. (2016) or BN-Inception Szegedy et al. (2015) backbones. We use the Recall@$k$ Jegou et al. (2011) and NMI Manning et al. (2010) (Normalized Mutual information) scores as our main metrics to compare our approach to the state-of-the-art methods, as summarized in Tab.2. We observe a significant boost in performance when applying our method on top of the conventional MS-Loss Wang et al. (2019) approach. Our approach can refine embeddings even starting from poor initial embeddings 4.

Table 3: **Ablation of different training losses** combined with our method. We provide R@1 scores on three main datasets *CUB200-2011*, *CARS196*, and *and SOP*. We repeat this evaluation using different pretrained models: MS-Loss model, Margin Loss , and ProxyAnchor

| Training Loss (R50) | CUB | Cars | SOP |
|---|---|---|---|
| MS-Loss | 67.5 | 87.8 | 77.4 |
| Ours (MS-Loss) | 73.2 | 90.9 | 81.8 |
| Margin[512] | 63.1 | 82.1 | 74.8 |
| Ours (Margin)[512] | 67.9 | 86.9 | 77.3 |
| ProxyAnchor[512] | 66.4 | 84.9 | 77.5 |
| Ours (ProxyAnchor)[512] | 69.8 | 88.5 | 80.2 |

## 4.3 OTHER BASELINE METHODS

Our method is highly versatile and can be effectively applied to a wide range of baseline approaches and networks, making it a valuable tool for many different applications. By leveraging initial embeddings of images computed using these baseline approaches, we are able to efficiently apply our method and achieve significant improvements in performance.

To test the effectiveness of our approach, we conducted experiments using multiple baseline approaches, including MS-Loss Wang et al. (2019), Margin Loss Wu et al. (2017) and ProxyAnchor loss Kim et al. (2020), and evaluated performance using the Recall@1 score on three datasets: CUB, Cars, and SOP. We provide the results of these experiments in Tab.3.

Our results demonstrate consistent and substantial improvement when using our method, indicating its potential to significantly enhance the performance by extracting information from the initial imperfect embeddings.

### 4.4 ITERATIVE IMPROVEMENT OF EMBEDDINGS

Our model for improving image embeddings takes an initial image embedding $e_q^0$ and its nearest neighbors as context $C_q$. We then iteratively apply cross-attention blocks to improve the embeddings. To understand how the quality of the embeddings change at each iteration, we take a trained model and compute the recalls at every level $e^t$, $t \in \{1, \ldots, 8\}$. We then plot the results, which can be seen in Fig. 3.

From the plot, we can see that there is a steady improvement in performance across all datasets, with each iteration resulting in better embeddings. It is worth noting that the biggest improvements in R@1 scores are seen in the earlier iterations. Therefore, our model demonstrates that by iteratively applying cross-attention blocks, we can improve the quality of image embeddings and achieve better performance across different datasets. This has an interesting implication that we can stop embeddings process at the earlier stage if we want to accelerate the computation.

### 4.5 MODEL ARCHITECTURE AND ABLATIONS

Our approach consists of two main components: the size of the neighborhood and the number of iteration steps, corresponding to the number of cross-attention blocks. To study the effect of these two factors, we trained different models using the same initial embeddings computed with the pretrained MS-Loss Wang et al. (2019) approach, but varying those two parameters. We observed consistent improvement in performance when increasing both components. It is worth noting that increasing only the number of cross-attention blocks while having only 2 neighbors in a neighborhood can even degrade the performance. We assume this is caused by fast overfitting exacerbated by the large number of parameters in cross-attention blocks. We also observed that going for a large number of cross-attention blocks or a large number of neighbors has diminishing returns. For that reason, we used a reasonably sized model with 8 cross-attention blocks and 8 nearest neighbors in a neighborhood. This is the model used when reporting scores in Tab. 2. More detailed ablation study results on this can be found in the Appendix.

We additionally ablated each component separately for our baseline model MS-Loss Wang et al. (2019). In Tab. 4, we first report the initial performance reached by the baseline model. Then, we add the cross-attention block, which has as its input for queries, keys and values the actual query image embedding $e_q$. We use 8 cross-attention blocks, which slightly improves the performance by making the embedding function more powerful. Alternatively, we add information about the nearest neighbors by averaging the embedding of a query point and 8 of its nearest neighbors. This does not affect or even degrades the performance. Finally, in the last line of each block of Tab. 4, we add both blocks - 8 nearest neighbors and 8 cross-attention blocks – and observe a significant improvement in performance over the baseline method.

### 4.6 VISUALIZATION OF ITERATIVE IMPROVEMENT

Our method produces not only the final embedding $e^T$, but also a sequence of intermediate embeddings $e^t \, \forall t \in \{1, \ldots, T\}$. We visualize intermediate embeddings to further prove the validity of our method.

First, we can show how the query embeddings $e_q$ change with respect to their nearest neighbors $NN(e_q)$. To do this, we project $e_q$ and its neighbors from a $d$-dimensional space into a 2-dimensional plane using PCA and t-SNE Maaten & Hinton (2008). Next, we optimize the position of the projected $e_q$ point on the 2D plane to match similarities between $NN(e_q)$ and $e_q$ in the original $d$-dimensional space. We visualize the embedding dynamics by projecting all $e_i^0$ onto a

Table 4: Ablation of the two main components of our method: information from the nearest neighbors, and cross-attention blocks used for aggregation. We report R@1 scores on three datasets. We evaluate different variants of model trained with MS-Loss.

| Variant ((R50) | CUB | Cars | SOP |
|---|---|---|---|
| MS-Loss | 67.5 | 87.8 | 77.4 |
| MS-Loss + CA | 68.6 | 88.9 | 78.1 |
| MS-Loss + NNs | 67.4 | 84.6 | 77.1 |
| Ours (CA + NNs) | 73.2 | 90.9 | 81.8 |

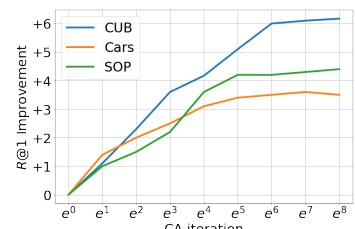

Figure 3: Relative improvement with respect to the initial embedding $e^0$ obtain from a vanilla MS-Loss approach. $e^t$ denotes the embedding level and $e^8$ stands for a final embedding of a trained model with 8 CA blocks.

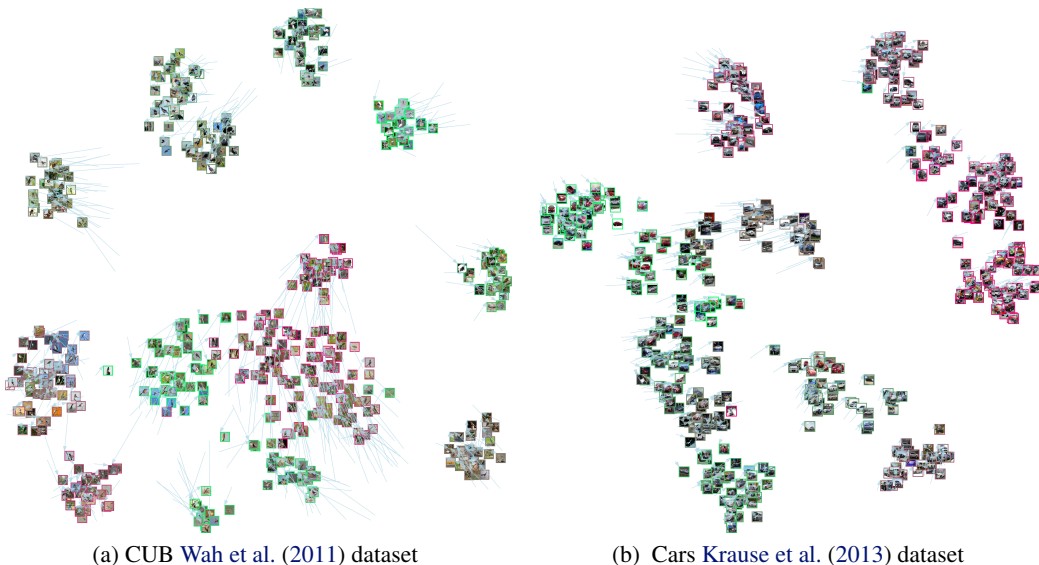

(a) CUB Wah et al. (2011) dataset             (b) Cars Krause et al. (2013) dataset

Figure 4: tSNE plot indicating movement between initial embedding and ultimate embedding. 500 images of each dataset are visualized. Best viewed on a monitor when zoomed in.

2-dimensional space using PCA and t-SNE, and then optimizing the positions of the 2D projections to match similarities $(e_i^T)^\top e_j$ of the original points in $d$-dimensional space at the final iteration. We repeat this visualization for the CUB and Cars datasets which can be found in Figure 4. We observe that not only are points with different labels (indicated in color frame) pushed apart, but we also observe a contracting behavior for points with the same label.

In addition, we provide more examples of the embedding dynamics in section B of our Appendix, where we show that query embeddings move in a direction where more embeddings with similar features exist, even if those neighboring embeddings were initially far away.

## 4.7 SPEED, MEMORY AND NUMBER OF PARAMETERS

The majority of the computational burden comes from the computations in the cross-attention blocks. This burden is twofold: an increase in the number of parameters and a reduction in inference speed.

Our model with 8 cross-attention blocks and embedding size of 512 dimensions contains 12M parameters. In comparison, conventional approaches mostly consist of the weights of the ResNet-50 He et al. (2016) backbone and weights of a projection head, this totals about 25M parameters. If we compare our model to a much larger baseline model, such as MS-Loss with the ResNet-101 backbone He et al. (2016), performance increases slightly, but the number of parameters is bigger

compared to our approach. R@1 scores are 71.8 and 86.4 on CUB and Cars, respectively, for the baseline MS-Loss Wang et al. (2019) with ResNet-101 backbone and they are lower than our scores of 73.2 and 90.9.

Regarding speed, our model can process 13K embeddings with nearest neighbors per second when the dataset of embeddings and nearest neighbors is provided. The most computationally heavy extra part is the computation of the matrix of all pairwise similarities, but it takes only 3.5 seconds on each split of the SOP dataset (each has around 60K images). In comparison, a baseline method like MS-Loss with the backbone encoder being ResNet-50 He et al. (2016), on images of size 224x224 and with embedding size 512, can only process 250 images per second. Thus, our approach, though having many parameters, adds a negligible computational overhead to the baseline approach that transforms an image $I_q$ into the initial embedding $e_q^0$. Speed is measured using the TitanXP GPU.

---

**Algorithm 1** Train

**Require:** $E, \phi$ - trained conventional DML approach, $b$ - batch size, $T$ - number of iterations, $k$ - neighborhood size. $\mathcal{I}$ - dataset with images and class labels
1: Compute all initial embeddings $e_i^0 := \phi(E(I_i)) \, \forall I_i \in \mathcal{I}$.
2: Precompute $k$ nearest neighbors as context $NN(e_i^0) \, \forall i$.
3: Initialize $\mathcal{E} := \{(e_i^0, NN(e_i^0))\}$
4: Initialize weights of $Q^t, K^t, V^t \, \forall t \in \{1, .., T\}$ of $CA^t$
5: **while** not converged **do**
6:     Sample $b$ pairs $(e_i^0, NN(e_i^0)) \in \mathcal{E}$
7:     **for** $\forall i \in \{1, .., b\}$ **do**
8:         $C_i := NN(e_i^0)$
9:         **for** $t = 1$ to $T$ **do**
10:             $e_i^t = e_i^{t-1} + \text{CA}^t(e_i^{t-1}, C_i, C_i)$
11:             Normalize $e_i^t = e_i^t / ||e_i^t||_2$
12:         **end for**
13:     **end for**
14:     Compute $\mathcal{L}$ with $e_i^T \, \forall i$
15:     Backprop from $\mathcal{L}$ into $\theta_{Q^t}, \theta_{K^t}, \theta_{V^t} \forall t$.
16: **end while**

**Algorithm 2** Inference

**Require:** $E, \phi$ - trained conventional DML approach, $T$ - number of iterations, $k$ - neighborhood size, $CA^t$ - trained cross-attention blocks, $\mathcal{I}$ - dataset with test images
1: **for** $\forall I_i \in \mathcal{I}$ **do**
2:     Compute all initial embeddings $e_i^0 := \phi(E(I_i))$.
3: **end for**
4: Precompute $k$ nearest neighbors as context $NN(e_i^0) \, \forall i$.
5: Initialize $\mathcal{E} := \{(e_i^0, NN(e_i^0))\}$
6: **for** $\forall i$ **do**
7:     $C_i := NN(e_i^0)$
8:     **for** $t = 1$ to $T$ **do**
9:         $e_i^t = e_i^{t-1} + \text{CA}^t(e_i^{t-1}, C_i, C_i)$
10:         Normalize $e_i^t = e_i^t / ||e_i^t||_2$
11:     **end for**
12: **end for**
13: Compute $s_{ij} = (e_i^T)^\top e_j^T \, \forall i, j$
14: Use $s_{ij}$ to retrieve nearest neighbors.

---

## 5 CONCLUSION

Our approach covers a gap between the conventional approaches to DML that have no access to the test distribution and the approaches utilizing self-supervised training or even partial labels information at the inference stage. It takes neighboring points into account when embedding an image. By using cross-attention to extract meaningful information from other samples, the method improves the local embedding of the image, allowing it to also better adapt to domain shifts common in DML. The proposed method outperforms the state of the art on common DML benchmark datasets. Our approach can be easily incorporated into existing DML methods at a negligible additional cost during inference, thus underlining its effectiveness and practicality.

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
