# ATTEND TO CONTEXT FOR REFINING EMBEDDINGS IN DEEP METRIC LEARNING
# – SUPPLEMENTARY MATERIALS –

## A    IMPLEMENTATION DETAILS

Our model consists of two parts: a baseline approach that transforms images $I_i$ into initial embeddings $e_i \in \mathbb{R}^d$, and our approach that transforms $e_i^0$ into $e_i^T$.

Initial image embeddings $e_i^0$ are obtained using the MS-Loss approach with images resized to $256 \times 256$ and then taking a central crop of size $224 \times 224$. Embeddings are of default size 512.

Our default model has 8 cross-attention blocks, and the neighborhood comprises 8 neighbors. All experiments are conducted on the TitanXP GPU. Our cross-attention blocks have just one head, as suggested in the original Perceiver paper Jaegle et al. (2021). We also use the SiLU activation function Elfwing et al. (2018) in the cross-attention projection layers. Skip connections are applied between the input and output of every cross-attention block. Additionally, we have a skip connection between the query projection head input and output. After every cross-attention block, we normalize the representation $e_i^t$.

For optimization, we use the Adam optimizer Kingma & Ba (2015) with a learning rate of $1e - 4$ and default $\beta_1$ and $\beta_2$ parameters. No learning rate scheduler is applied. The batch size is 128 for all experiments, and the model is implemented using the Tensorflow2 framework.

## B    ADDITIONAL HYPER PARAMETERS

In the main script, we define the Multi-Similarity loss using values of $\alpha = 2$, $\beta = 40$, and $\lambda = 0.5$ in Eq. 4. During batch training, we only sample 2 positive samples per batch.

Our method can work with embeddings of arbitrary sizes. Specifically, we can take embeddings of dimensionality $d_1$ as output from a baseline method, feed them into our chain of cross-attention blocks, and apply a new linear layer + $l_2$ normalization layer to obtain embeddings of dimensionality $d_2$.

## C    ADDITIONAL VISUALIZATIONS

We additionally visualize a few small neighborhoods for three datasets used throughout our paper and observe similar clustering behavior. Additionally, we use PCA + tSNE Maaten & Hinton (2008) projections to visualize large groups of points and show how they evolve over time, specifically embeddings $e^0$ to $e^8$. To illustrate the movement of the embeddings, we interpolate between each of the eight embeddings. These drifts of embeddings are visualized in the .mp4 files attached to the supplementary material. These experiments are conducted on three primary datasets: CUB Wah et al. (2011), Cars Krause et al. (2013), and SOP Oh Song et al. (2016). See Fig.2, Fig.3, Fig.4, Fig.7, Fig.8, and Fig.9 for illustrations.

## REFERENCES

Stefan Elfwing, Eiji Uchibe, and Kenji Doya. Sigmoid-weighted linear units for neural network function approximation in reinforcement learning. *Neural networks*, 2018.

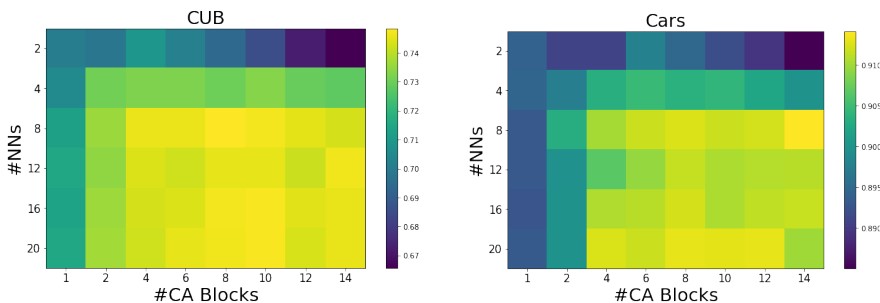

Figure 1: Retrieval performance depends on the number of nearest neighbors used for evaluation and the number of cross-attention blocks. We report R@1 score.

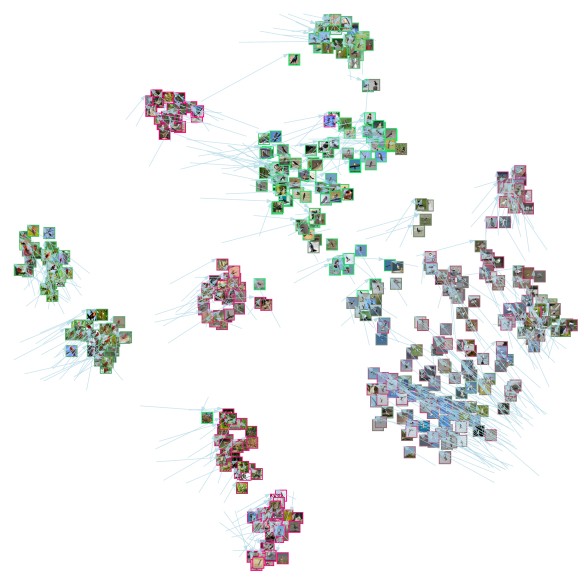

Figure 2: tSNE plot indicating movement between initial embedding and ultimate embedding. 500 images of the CUB Wah et al. (2011) dataset are visualized. Best viewed on a monitor when zoomed in.

Andrew Jaegle, Felix Gimeno, Andrew Brock, Andrew Zisserman, Oriol Vinyals, and João Carreira. Perceiver: General perception with iterative attention. In *ICML*, 2021.

Diederik P Kingma and Jimmy Ba. Adam: A method for stochastic optimization. In *ICLR*, 2015.

Jonathan Krause, Michael Stark, Jia Deng, and Li Fei-Fei. 3d object representations for fine-grained categorization. In *Proceedings of the IEEE International Conference on Computer Vision Workshops*, 2013.

Laurens van der Maaten and Geoffrey Hinton. Visualizing data using t-sne. *JMLR*, 2008.

Hyun Oh Song, Yu Xiang, Stefanie Jegelka, and Silvio Savarese. Deep metric learning via lifted structured feature embedding. In *Proceedings of the IEEE Conference on Computer Vision and Pattern Recognition*, 2016.

C. Wah, S. Branson, P. Welinder, P. Perona, and S. Belongie. The Caltech-UCSD Birds-200-2011 Dataset. Technical report, California Institute of Technology, 2011.

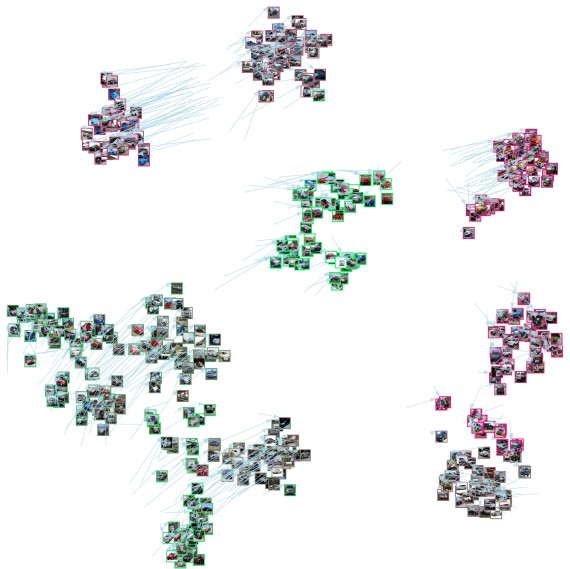

Figure 3: tSNE plot indicating movement between initial embedding and ultimate embedding. 500 images of the Cars Krause et al. (2013) dataset are visualized. Best viewed on a monitor when zoomed in.

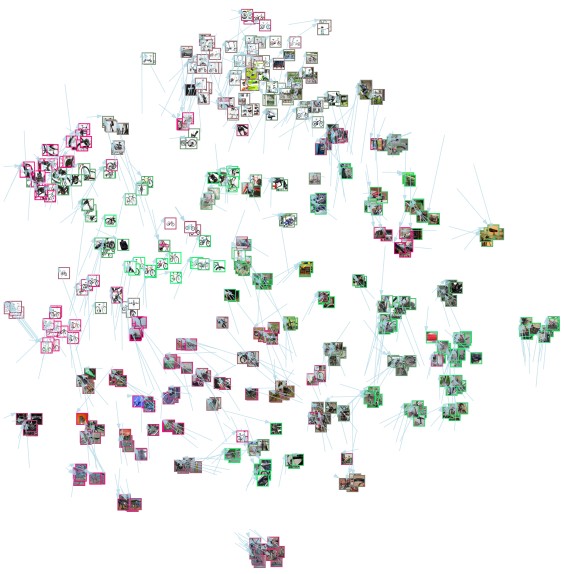

Figure 4: tSNE plot indicating movement between initial embedding and ultimate embedding. 500 images of the SOP Oh Song et al. (2016) dataset are visualized. Best viewed on a monitor when zoomed in.

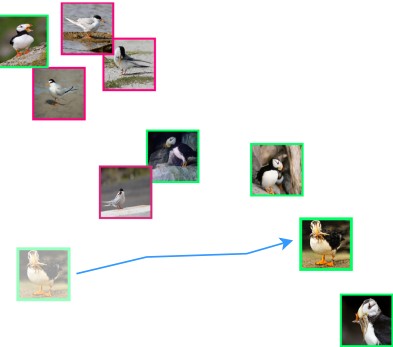

Figure 5: tSNE plot indicating movement of a query image between initial embedding and ultimate embedding. 8 NNs are shown. Frame color specifies label. Images from the CUB Wah et al. (2011) dataset are impainted.

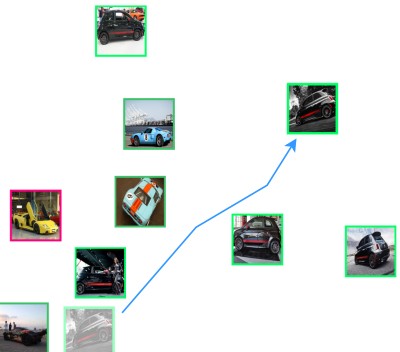

Figure 6: tSNE plot indicating movement of a query image between initial embedding and ultimate embedding. 8 NNs are shown. Frame color specifies label. Images from the Cars Krause et al. (2013) dataset are impainted.

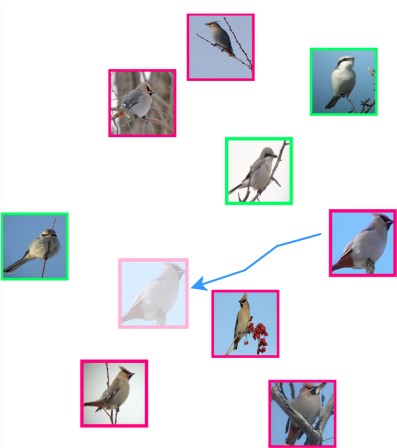

Figure 7: tSNE plot indicating movement of a query image between initial embedding and ultimate embedding. 8 NNs are shown. Frame color specifies label. Images from the CUB Wah et al. (2011) dataset are impainted.

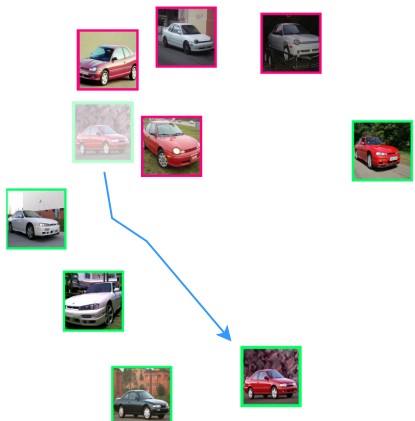

Figure 8: tSNE plot indicating movement of a query image between initial embedding and ultimate embedding. 8 NNs are shown. Frame color specifies label. Images from the Cars Krause et al. (2013) dataset are impainted.

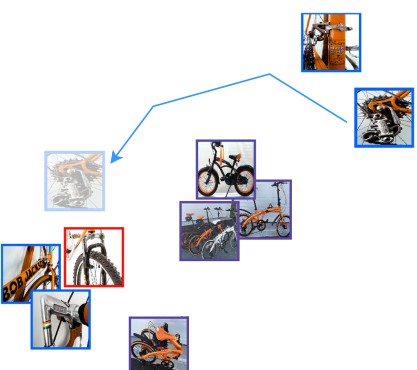

Figure 9: tSNE plot indicating movement of a query image between initial embedding and ultimate embedding. 8 NNs are shown. Frame color specifies label. Images from the SOP Oh Song et al. (2016) dataset are impainted.