# OpenReview forum: "Attend to Context for Refining Embeddings in Deep Metric Learning"
_ICLR.cc/2024/Conference — ICLR 2024 Conference Withdrawn Submission_

### Official Review · Reviewer_DDGY · 2023-10-29

**Soundness:** 1 poor
**Presentation:** 1 poor
**Contribution:** 1 poor
**Rating:** 1
**Confidence:** 5

**Summary:**

This paper addresses the DML problem using k-nearest neighbors. Cross-attention is applied to incorporate meaningful information from other samples. Experiments are conducted on the DML benchmarks to validate the effectiveness of the proposed methods.

**Strengths:**

I couldn't identify any significant strengths in this paper. Although I initially only intended to read the abstract, I continued to read the introduction in order to provide more detailed feedback in my comments.

**Weaknesses:**

1.The writing is poor.

2.The authors lack an in-depth background survey. For example, the statement "conventional deep metric learning approaches typically process each image independently of others" is not sufficiently supported by references. Previous works such as [r1] and [r2] have proposed solutions to address this problem using Graph Neural Networks (GNN) and message passing networks, respectively. The authors did not mention these prior works, which are important for providing context and understanding the existing solutions in deep metric learning.

[r1] Kan, Shichao, et al. "Local semantic correlation modeling over graph neural networks for deep feature embedding and image retrieval." IEEE Transactions on Image Processing 31 (2022): 2988-3003.

[r2] Seidenschwarz, Jenny Denise, Ismail Elezi, and Laura Leal-Taixé. "Learning intra-batch connections for deep metric learning." International Conference on Machine Learning. PMLR, 2021.

3. It's important for the authors to provide a fair comparison with state-of-the-art methods in deep metric learning to establish the competitiveness of their proposed approach. Reviewing the DML works of Sungyeon Kim et al., Karsten Roth et al., Shichao Kan et al., Xinshao Wang et al., and citing relevant state-of-the-art methods in their paper would strengthen the comparison and provide a more accurate assessment of this approach.

**Questions:**

See the weaknesses.

---

### Official Review · Reviewer_haCf · 2023-10-31

**Soundness:** 3 good
**Presentation:** 3 good
**Contribution:** 2 fair
**Rating:** 5
**Confidence:** 4

**Summary:**

The paper addresses the deep metric learning problem. Given an off-the-shelf network, embedding, the paper proposed to enhance the embedding of the query using the k-nearest neighbor in the gallery set. Specifically, it learns a cross-attention network, which takes the original embedding and its k-nearest neighbors as input and outputs a new embedding as the final embedding for the query. The paper evaluated the proposed method on multiple benchmarks including CUB, Cars and SOP. All show good performance gain and achieve the SOTA performance.

**Strengths:**

The paper is mainly clear and easy to follow. The idea of learning a network to enhance the query feature via kNN of the query feature seems novel and interesting.

The proposed method can be applied to multiple baseline networks. The paper shows how the proposed method enhances MS-Loss, Margin Loss and ProxyAnchor loss. The proposed method achieves the SOTA performance when applied to MS-Loss, a widely used deep metric learning loss.

**Weaknesses:**

I have concerns about several contribution statements in the paper.

1. “The proposed method adds negligible computation overhead at inference time”. It depends on the size of the dataset. It might be true for the listed benchmarks which have only <100k images at inference time. However, for a million or even billion-scale dataset, finding k-nearest neighbors is non-trivial. A sub-linear indexing (like LSH or Product-Quantization) may be needed. Also, sub-linear indexing may affect the accuracy of finding the kNN thus affect the quality of the final embeddings.

2. “Targets the problem of a distribution shift in DML”. I didn’t find any discussion either in the method section or the experiment section. It’s not clear how the proposed method can help mitigate the distribution shift. Also, there is no experiment explicitly showing that the distribution shift got mitigated (except for the overall performance).

3. “Breaks the assumption of conventional approaches that images exist independently from each other in the embedding space”. I don’t fully agree with this. It’s true that in the benchmark, all queries have corresponding samples in the dataset which have the same category label. However, it may not be true in other problems where out-of-domain queries may exist. The proposed method to me, is more like “retrieval-augmented” feature enhancement. It aligns better for Retrieval-Augmented-Generation (RAG) instead of DML. The text set in DML is a carefully crafted dataset for evaluating the DML feature extractor (all categories have almost the same number of images, not like the long-tail distribution in real-world problem). In my opinion, it’s not designed as an external knowledge base to enhance the query feature. It’s not clear the proposed method works or not with a large-scale common knowledge base. The inference time may also be a problem when the knowledge base becomes large.

4. The main method introduced by the paper is the cross-attention model. However, if I understand correctly, any sequence model (for example transformer) should work for the context modeling part. For example, each feature, including the original query and it’s kNNs can be considered as the input embedding of a sequence, and the goal is just to get a new output embedding. In the paper, the ablation study only conducts on simple averaging and cross-attention with different number of blocks, no other sequence models are compared.

**Questions:**

From technical point of view, the paper is easy to understand. The performance of the proposed method is also significant. However, my major concern is the setting of the problem which uses the evaluation set as a knowledge base to enhance the query feature. I’d like to discuss it with the authors and other reviewers in the rebuttal period.

**Details Of Ethics Concerns:**

I don't think there exists ethics concern for this submission.

---

### Official Review · Reviewer_6eUw · 2023-11-03

**Soundness:** 1 poor
**Presentation:** 2 fair
**Contribution:** 2 fair
**Rating:** 3
**Confidence:** 5

**Summary:**

This paper proposes a test-time adaptation methods which can adapt the embeddings to local embeddings at the test time. They leverage the local geometry of nearest neighbors to improve the learned embeddings without fine-tuning the metric network. Experiments on the widely used three deep metric learning benchmarks verify the effectiveness of the proposed method.

**Strengths:**

1. The motivation of this paper makes sense. The distribution shift is a long-standing problem in deep metric learning, especially in the adopted zero-shot setting.
2. The performance is strong. It can improve existing methods and achieves good results

**Weaknesses:**

1. Lack of novelty. How to obtain better retrieval or clustering performance using test samples is widely explored in the area of person re-identification and face recognition, such as re-rank. It has been shown that utilizing test-time information would lead performance improvement, so the results of this paper do not surprise me.
2. Lack of technical contributions. The proposed method is basically using the cross-attention mechanism to incorporate information from other samples.
3. Lack of experiments. Firstly, the comparisons with SOTA methods are not fair. The compared methods do not use test samples for further refinement, and do not use additional layers such as cross-attention to obtain the embeddings. Secondly, a lot of recent related works are missing, such as [1]. Thirdly, the authors do not compare their methods with other methods that leverage test-time samples.

[1] Wang C, Zheng W, Zhu Z, et al. Introspective deep metric learning[J]. IEEE Transactions on Pattern Analysis and Machine Intelligence, 2023.

**Questions:**

See weakness.

---

### Official Review · Reviewer_NTPu · 2023-11-08

**Soundness:** 2 fair
**Presentation:** 3 good
**Contribution:** 3 good
**Rating:** 6
**Confidence:** 3

**Summary:**

This paper proposes a neighbourhood-aware method for enhancing the performance of deep metric leaning model. It can be incorporated into many existing DML approaches.

**Strengths:**

1) Enables neighbourhood-aware embeddings;
2) has the potential of dealing with distribution shift;
3) can have a wide application In DML as a flexible module.

**Weaknesses:**

1) High complexity - Retrieval of neighbours and storing all embeddings can be computationally prohibitive.
2) the claim on dealing with distribution shift is not supported in any form (unless I have missed something?)
3) I am not sure if it is appropriate to say "conventional deep metric learning approaches typically process each image independently of others". They embed images interactively in training, through e.g., triplet loss. In testing the process are mostly independent, due to lack of access to training embeddings.

Incomplete sentence: "And this neighbourhood always has a correct retrieval sample which may not be the nearest neighbour to our query sample"

**Questions:**

How would the backbone model perform given same time/space budget compared to the proposed model? For example, add more layers or attach other modules to the backbone model.